

# Preliminary study on the association between lignan metabolites and CT non-destructive testing of coconut fruit at different developmental stages

Chengxu Sun[1,*], Xuejing Ma[1,2,*], JeromeJeyakumar John Martin[1], Hongxing Cao[1], Yu Zhang[3], Yanming Gao[2], Chunyu Xing[4] and Mingming Hou[1,5]

[1] Coconut Research Institute, Chinese Academy of Tropical Agricultural Sciences/Hainan Key Laboratory of Tropical Oil Crops Biology, Wenchang, Hainan, China
[2] Horticulture Laboratory, School of Wine and Horticulture, Ningxia University, Ningxia, China
[3] School of Computer Science and Technology, Hainan University, Haikou, Hainan, China
[4] School of Food Engineering, Tianjin Tianshi College, Tianjin, China
[5] College of Life Sciences, Henan University, Kaifeng, China
* These authors contributed equally to this work.

Corresponding author
Hongxing Cao,
hongxing1976@163.com

## ABSTRACT

Lignans play a crucial role in maintaining plant growth, development, metabolism and stress resistance. Computed tomography (CT) imaging technology can be used to explore the internal structure and morphology of plants, and understanding the correlation between the two is highly significant. In this study, the content of lignan metabolites in coconut water was determined using liquid chromatography. The internal structure data of coconut fruit was obtained by CT scanning, and the relationship between lignan metabolites and CT image data at different developmental stages was evaluated using partial least square (PLS) regression. The results showed that the total lignan content in coconut water initially decreased, then increased, and gradually decreased after the maturity stage. The Wenye No. 5 variety exhibited higher levels of Epiturinol, Turbinol, Isobarinin-9′-o-glucoside, 5′-methoxy-rohanoside, Rohan rosin-4,4′-di-o-glucoside, turbinol-4-O-glucoside, cycloisoperinolin-4-O-glucoside compared to local coconuts. Coconut meat had the greatest effect on Rohan rosin-4,4′-di-o-glucoside, coconut water on Daphne, and coconut shell and coconut fiber on Larinin-4′-o-glucoside. The data from different parts of coconut fruit's images showed a significant correlation with the content of lignan metabolites. This study has preliminarily explored the correlation between non-destructive testing of coconut fruit and its development process of coconut fruit, providing a new approach and method for further research on non-destructive testing of coconut fruit development.

## INTRODUCTION

Coconut (*Cocos nucifera* L.), a typical tropical cash crop, is known as the "tree of life" (*Lédo Ada et al., 2019*; *Sun et al., 2019*; *Sun et al., 2014*). Fruits are recognized for their nutritional value due to their beneficial components, such as vitamins, sugars, antioxidants, minerals and acids. Because of these important nutritional components, coconut fruits play a vital role in a healthy human diet, although nutritional value varies depending on the type of fruit, cultivar and maturity stage.

Coconut water is one of the most popular natural beverages. However, a relatively overlooked aspect is the presence of lignin compounds in coconut water. Lignan are metabolites that significantly contribute to the nutritional value of coconut water (*Wang et al., 2008*). Lignans are natural phenolic compounds (*Wang et al., 2017*; *Tao et al., 2017*), support plant cell walls and play a crucial role in plant growth and development (*She et al., 2017*; *Cesarino, 2019*). For example, Epiturinol, a less commonly known derivative of turbinol, potentially possesses unique biochemical properties significant in plant metabolism and stress responses (*Grotewold, 2005*). Another compound, Rohan rosin phenol-4′-O-glucoside is a phenolic glucoside derived from rosin, a natural resin from pine trees. The glucoside form impacts its biological properties, including roles in plant defense and resilience against environmental stress (*Lattanzio, Lattanzio & Cardinali, 2006*). Lignans are natural phytoestrogens with with various pharmacological activities, including anti-tumor activity, anti-ultraviolet radiation, central nervous system regulation, and antioxidant properties (*Tu et al., 2014*; *Grabber et al., 2004*). For instance, 5′-Methoxy-rohanoside, a methoxy derivative of rohanoside, includes a methoxy group that influences its pharmacokinetic and pharmacodynamic properties, affecting various biochemical pathways in plants (*Wink, 2003*). Lignan compounds, found in plants like forsythia and sesame, have been extensively researched for their anti-inflammatory, antioxidant, and sedative properties (*Qi et al., 2021*; *Hu & Qin, 2019*; *Yan et al., 2019*). Additionally, turbinol a naturally occurring compound in certain plant species, known for its potential therapeutic applications and biological activities, including antioxidative and antimicrobial properties, which play a role in plant defense mechanisms against pathogens (*Dixon, 2001*). Computer tomography (CT) nondestructive testing is an advanced testing technology with promising application in plant detection (*Tollner et al., 1992*; *Barcelon, Tojo & Watanabe, 1999*). CT scan with synchrotron radiation can not only detect CT values of various parts of coconut, but also offer several advantages such as high X-ray flux, high-speed reconstruction algorithm and high-speed detectors. This technology allows for the generation of 3D and high-resolution images in a short period, with high data acquisition.

The purpose of this study is to establish a model of lignan metabolites for coconut fruit development and to introduce a novel concept for non-destructive testing of coconut fruit development. Initially, the study focuses on the development of lignan metabolites at various stages of coconut fruit maturation. Recent advancements in CT nondestructive testing of coconuts have been made (*Zhang et al., 2023*). However, the relationship

between the lignan metabolite content of coconut fruit at various developmental stages and the CT image data remains unknown. Therefore, the present study aims to investigate the correlation between metabolite content and CT non-destructive testing of coconut fruit, laying a foundation for the further development and cultivation of superior coconut varieties with high lignan content.

# MATERIALS AND METHODS

## Material selection

The experimental materials were collected from 10-year-old Wenye No. 5 and local coconut (CK) trees at the research base of the Coconut Research Institute, Chinese Academy of Tropical Agricultural Sciences, as shown in Fig. S1. The characteristics of Wenye No. 5 are as follows: vigorous tree growth, vertical tree posture, flowering and fruiting 3 to 4 years after planting, high yield period 7 years after planting, natural life span of approximately 60 years, and economic life span of approximately 35 years. This variety is monoecious, with overlapping flowering, self-pollination, and annual flowering. The fruit is small and round, weighing of 0.55 kg, and has a high yields, with an average plant yield of 140 and a maximum yield of over 300. Coconuts were harvested at various developmental stages (2, 4, 6, 8, 10 and 12 months), and 100 ml of coconut water was taken from each. The samples were frozen in liquid nitrogen and stored at $-80\,^{\circ}\mathrm{C}$ for analysis. The samples were repeated three times.

## Test equipment and method

This study employed a dual-source CT scanner (SOMATOM Definition Flash, Siemens Healthineers, Erlangen, Germany) to acquire images of the coconut samples. The CT scanning parameters were set as follows: section thickness/increment = 0.6 mm/75%, tube voltage = 120 kv, tube current = 250 mAs, field of view = 400 × 400 mm, and frame rate = 0.5 s/rev. The contents of lignan metabolites were determined by liquid chromatography. The mobile phase consisted of solvent A (pure water with 0.1% formic acid), and solvent B (acetonitrile with 0.1% formic acid). Sample measurements were performed using a gradient program with an initial composition of 95% solvent A and 5% solvent B. The image parameters were obtained through CT scanning. The recorded results of coconut CT scan data are presented in Table S1, and the scanning results of images at different times are shown in Fig. S2.

## Determination and analysis of lignan metabolome

The combination of chromatography and mass spectrometry facilitates the entire process, from the separation of substances by chromatography to their identification by mass spectrometry. Lignans in coconut water can be accurately measured both qualitatively and quantitatively using ultra-high performance liquid chromatography tandem mass spectrometry (UHPLC-MS/MS).

After thawing the samples from the $-80\,^{\circ}\mathrm{C}$ freezer, swirl for 10 s to ensure thorough mixing. A 9 mL aliquot of the mixed sample was then transferred to a 50 mL centrifuge

tube and immersed in liquid nitrogen. Once the sample was completely freeze dried, 300 uL 70% methanol internal standard extraction solution was added, and the mixture was swirled for 3 min. The mixture was then centrifuged at 12,000 rpm and 4 °C for 10 min. The resulting supernatant was filtered through a microporous filter membrane (0.22 μm) and stored in the sample vial for UPLC-MS/MS testing.

The metabolite data obtained from mass spectrometry were preprocessed and quality controlled to ensure accurate quantification. Differential metabolites were screened using orthogonal partial least-squares discrimination analysis (OPLS-DA) and differential multiple integration. The screening thresholds were VIP ≥ 1, Fold_Change ≥ 2 or Fold_Change ≤ 0.5 to identify differential metabolites.

The data acquisition instrument system consisted of Ultra Performance Liquid Chromatography (UPLC) (Nexera X2; Shimadzu, Kyoto, Japan) and Tandem mass spectrometry (4500 QTRAP; Applied Biosystems, Waltham, MA, USA). The general workflow of mass spectrometry is shown in Fig. S3.

The liquid phase conditions were as follows:

1) Column: Agilent SB-C18 1.8 μm, 2.1 mm * 100 mm;

2) Mobile phase: Phase A is ultra-pure water with 0.1% formic acid; phase B is acetonitrile with 0.1% formic acid.

3) Elution gradient: The proportion of phase B started at 5% at 0.00 min, increases linearly to 95% within 9.00 min, and maintained at 95% for 1 min. From 10.00–11.10 min, the proportion of B phase decreases to 5%, and was balanced at 5% until 14 min.

4) Flow rate 0.35 mL/min; column temperature: 40 °C; sample size: 4 μL.

The mass spectrometry conditions were as follows:

Linear Ion Trap (LIT) and Triple Quadrupole (QQQ) scanning were performed using the Triple quadrupole linear ion TRAP Mass Spectrometer (QTRAP), AB4500 Q TRAP UPLC/MS/MS system equipped with an ESI Turbo ion spray interface. The Analyst 1.6.3 software (AB Sciex, Framingham, MA, USA) was used to control both positive and negative ion modes. The optimized ESI source operating parameters were optimized as follows: turbo spray ion source, source temperature of 550 °C, and ion spray voltage of 5,500 V in positive ion mode and −4,500 V in negative ion mode. The ion source gases were set to 50 psi for gas I (GSI), 60 psi for gas II (GSII), and 25.0 psi for the curtain gas (CUR). The collision-induced ionization parameter was set to high. Prior to analysis, the instrument was tuned and calibrated using 10 and 100 μmol/L polypropylene glycol solutions in QQQ and LIT modes, respectively. The QQQ scan used multiple reaction monitoring (MRM) mode with the collision gas (nitrogen) set to medium. Through further optimization, the declustering potential (DP) and collision energy (CE) for each MRM ion pair were determined. A specific set of MRM ion pairs was monitored during each period based on the metabolites elution times. The multi-modal graph of MRM metabolite detection is shown in Fig. S4.

## Determination of CT values of different structures of coconut fruit at different development stages

The changes in CT values for different structures over time were measured using a method described by Zhang et al. (2023).

The calculation formula for CT value is:

CT value = ((Ux-U water at 73 KV)/Uwater at 73 KV)*K.

Note: K is a constant with a value of 1,000. Ux represents the attenuation coefficient of the substance, Uwater is the attenuation coefficient of water, (typically referred to as the Hounsfield unit HU), Uair = −1,000.

## Data and statistical analysis

In this experiment, each measurement was triplicated to ensure accuracy. The resulting data were expressed as the mean value ± standard deviation (SD), with a sample size of three ($n = 3$). To determine the differences between treatments, we employed one-way ANOVA, followed by Duncan's test to identify significant differences among means at probability levels of 0.05 and 0.01 using SPSS software (version 19.0, IBM, Armonk, NY, USA). Additionally, we performed correlation analysis, principal component analysis, and cluster analysis using Data Processing System (DPS, version 9.01) to further explore the data (Sun et al., 2021).

# RESULTS

## Analysis of lignan metabolome data

### Analysis of lignan metabolite content changes in coconut water at different developmental stages

The content of lignan metabolites in local coconut water was highest at 2, 10, 12 months of fruit age (Fig. 1). Specifically, Cephaloethin was present at high levels only at 2 months of fruit age. In contrast, Epiturinol, Turbinol, turbinol-4-O-glucoside, and Rohan rosin phenol-4′-O-glucoside had the highest levels of metabolites at 10 months. The contents of 5′-methoxy-rohanoside, Isobarinin-9′-o-glucoside and Larinin-4′-o-glucoside were highest at 12 months.

The content of lignan metabolites in the water of Wenye No. 5 was higher at 2 and 12 months, (Fig. 2). The content of Epiturinol, Turbinol,Isobarinin-9′-O-glucoside, Dihydrodehydrodicarboxylate-4-O-glucoside, turbinol-4-O-glucoside were highest and most prominent at 10 months. In contrast, Cephaloethin and Rohan rosin phenol-4′-O-glucoside had high content at 2 months. Notably, the contents of 5′-methoxy-rohanoside and cycloisoperinolin-4-O-glucoside were highest only at 12 months.

According to the dynamic changes of lignan metabolite accumulation (Fig. 3), the lignan content of Wenye No. 5 was lower than that of local coconut only at 6 and 8 months, but higher at the other stages of development. The change in lignan content of Wenye No. 5 from 4 to 6 months of fruit age was stable and similar to that of local coconut.

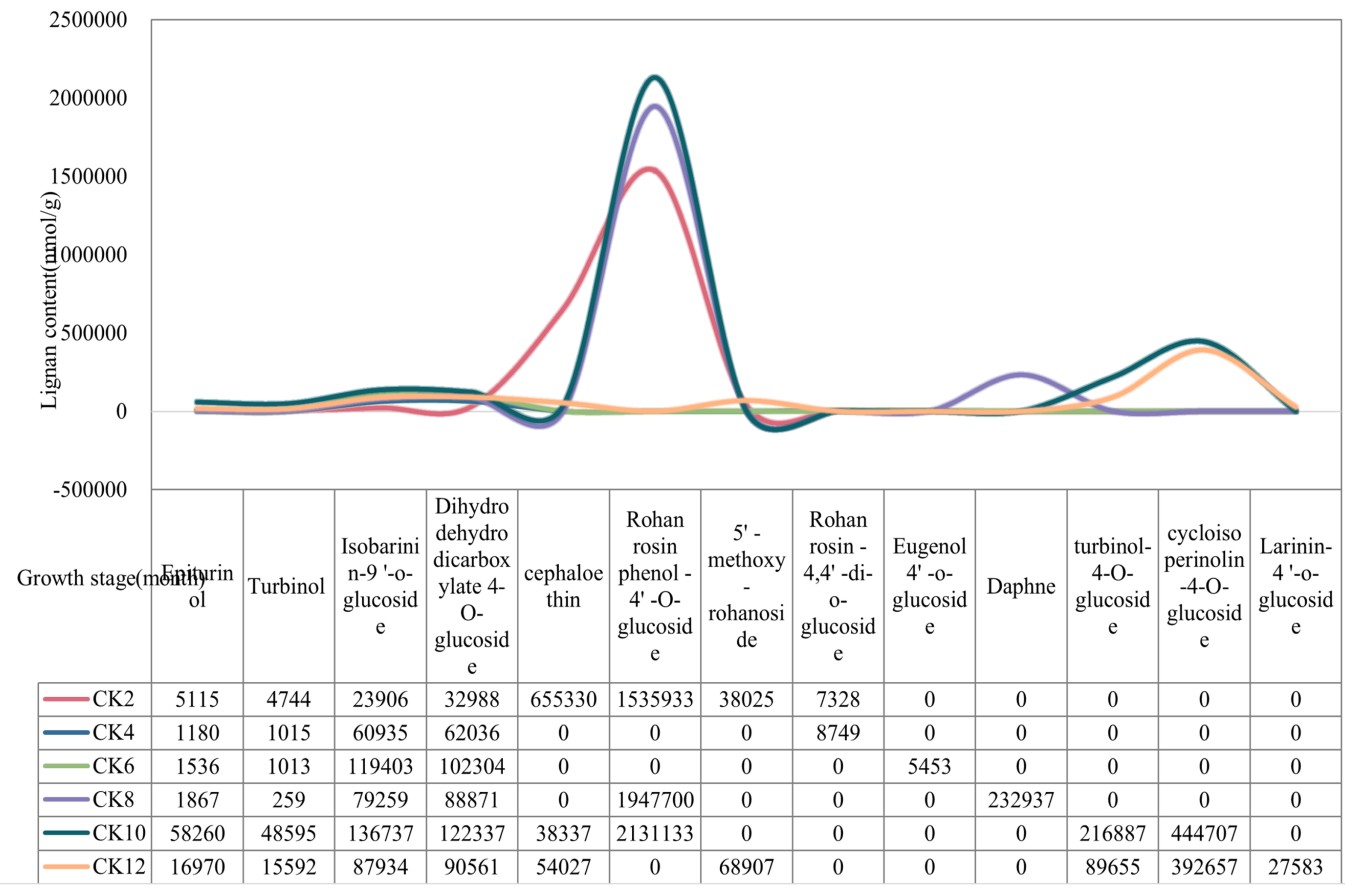

| Growth stage(month) | Epiturinol | Turbinol | Isobarinin-9'-o-glucoside | Dihydro dehydro dicarboxylate 4-O-glucoside | cephaloethin | Rohan rosin phenol-4'-O-glucoside | 5'-methoxy-rohanoside | Rohan rosin-4,4'-di-o-glucoside | Eugenol 4'-o-glucoside | Daphne | turbinol-4-O-glucoside | cycloiso perinolin-4-O-glucoside | Larinin-4'-o-glucoside |
|---|---|---|---|---|---|---|---|---|---|---|---|---|---|
| CK2 | 5115 | 4744 | 23906 | 32988 | 655330 | 1535933 | 38025 | 7328 | 0 | 0 | 0 | 0 | 0 |
| CK4 | 1180 | 1015 | 60935 | 62036 | 0 | 0 | 0 | 8749 | 0 | 0 | 0 | 0 | 0 |
| CK6 | 1536 | 1013 | 119403 | 102304 | 0 | 0 | 0 | 0 | 5453 | 0 | 0 | 0 | 0 |
| CK8 | 1867 | 259 | 79259 | 88871 | 0 | 1947700 | 0 | 0 | 0 | 232937 | 0 | 0 | 0 |
| CK10 | 58260 | 48595 | 136737 | 122337 | 38337 | 2131133 | 0 | 0 | 0 | 0 | 216887 | 444707 | 0 |
| CK12 | 16970 | 15592 | 87934 | 90561 | 54027 | 0 | 68907 | 0 | 0 | 0 | 89655 | 392657 | 27583 |

**Figure 1 Changes in lignan content of local coconuts at different developmental stages.**

Notably, the lignan content of local coconut peaked at 10 months of fruit age (31,96,993 nmol/g), whereas Wenye No. 5 had its highest content at 2 months of fruit age (44,23,499 nmol/g), which was 1.38 times that of local coconut.

Figure 4 illustrates that three measurements of the same variety are taken simultaneously, indicating points that are relatively close together. However, for the two varieties of coconuts, the metabolites of local coconuts with fruit ages 10–12 months showed significant differences from Wenye No. 5. The metabolite difference between local coconut and Wenye No. 5 at 2 months was mainly reflected in pc2, and the variable with the greatest contribution to pc2 was 5′-methoxy-rohanoside. Therefore, this metabolite may be the main factor leading to the difference between the two varieties at 2 months. Secondary factors Cephaloethin and Larinin-4′-o-glucoside contributed to the difference in fruit age between two varieties at 2 months, while main factors Epiturinol and Turbinol caused difference in fruit age at 10 months. The study findings demonstrate significant relationships between various compounds. Specifically, a strong negative correlation between Eugenol 4′-o-glucoside and 5′-methoxy-rohanoside. Additionally, a strong

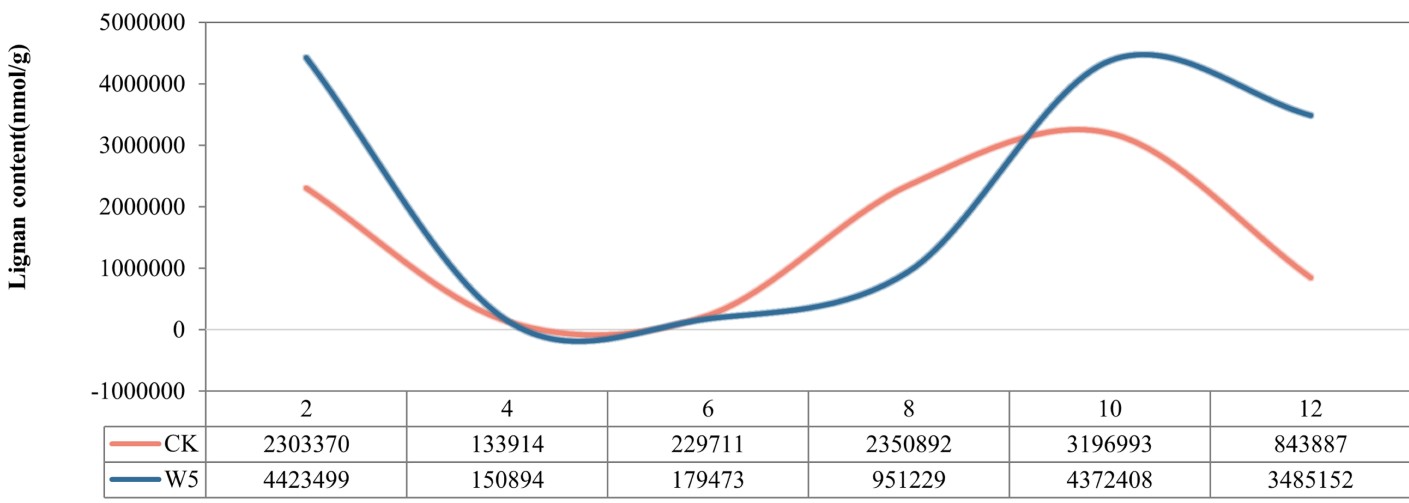

**Figure 2** The lignan content of Wenye No. 5 at different developmental stages.

| Growth stage(month) | Epiturinol | Turbinol | Isobarinin-9 '-o-glucoside | Dihydro dehydro dicarboxylate 4-O-glucoside | cephaloethin | Rohan rosin phenol - 4' -O-glucoside | 5' - methoxy - rohanoside | Rohan rosin - 4,4' -di-o-glucoside | Eugenol 4' -o-glucoside | Daphne | turbinol-4-O-glucoside | cycloisoperinolin -4-O-glucoside | Larinin-4 '-o-glucoside |
|---|---|---|---|---|---|---|---|---|---|---|---|---|---|
| W5-2 | 3103 | 2672 | 65628 | 53749 | 320570 | 3835633 | 124090 | 18053 | 0 | 0 | 0 | 0 | 0 |
| W5-4 | 1517 | 1798 | 74161 | 51113 | 0 | 0 | 0 | 22305 | 0 | 0 | 0 | 0 | 0 |
| W5-6 | 6833 | 6532 | 83422 | 80656 | 0 | 0 | 0 | 0 | 2029 | 0 | 0 | 0 | 0 |
| W5-8 | 33516 | 27420 | 127960 | 145230 | 0 | 578270 | 0 | 0 | 0 | 38834 | 0 | 0 | 0 |
| W5-10 | 107080 | 96335 | 1092653 | 1096640 | 11043 | 387607 | 0 | 0 | 0 | 0 | 656740 | 924310 | 0 |
| W5-12 | 39820 | 36679 | 585947 | 511413 | 24665 | 0 | 149553 | 0 | 0 | 0 | 312953 | 1749167 | 74955 |

| Growth stage(month) | 2 | 4 | 6 | 8 | 10 | 12 |
|---|---|---|---|---|---|---|
| CK | 2303370 | 133914 | 229711 | 2350892 | 3196993 | 843887 |
| W5 | 4423499 | 150894 | 179473 | 951229 | 4372408 | 3485152 |

**Figure 3** Lignan content in different developmental stages of coconut.

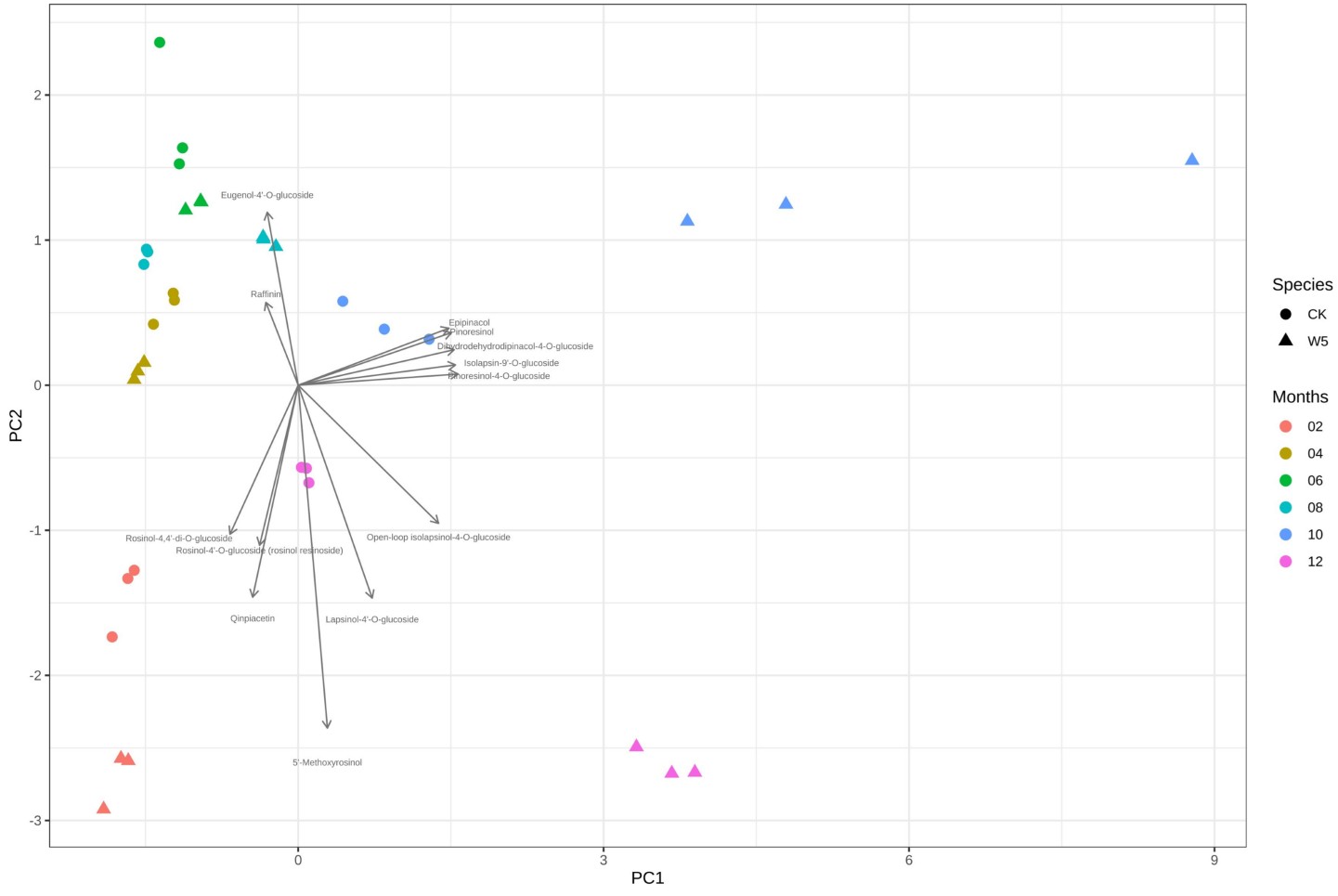

**Figure 4 Analysis on the difference of lignans in different coconut varieties at different developmental stages.**

positive correlation was observed between Epiturinol and Turbinol. Furthermore, a weak correlation was identified between Daphnetin and Turbinol.

### Analysis of lignan metabolites difference between two coconut varieties at different developmental stages

Differential metabolite statistics provide valuable insights into the variations between different coconut varieties, such as local coconut and Wenye No. 5. Table S2 shows the number of statistically different metabolites, which analyze the differences in metabolite composition and concentrations between the two varieties. By comparing the metabolites, we can identify unique compounds or varying levels of certain metabolites. The Fig. 5A reveals changes in metabolite numbers at different developmental stages, with four differential metabolites up-regulated and one down-regulated in the 2 *vs*. 4 control group. In contrast, the 10 *vs*. 12 group has highest number of up regulated differential
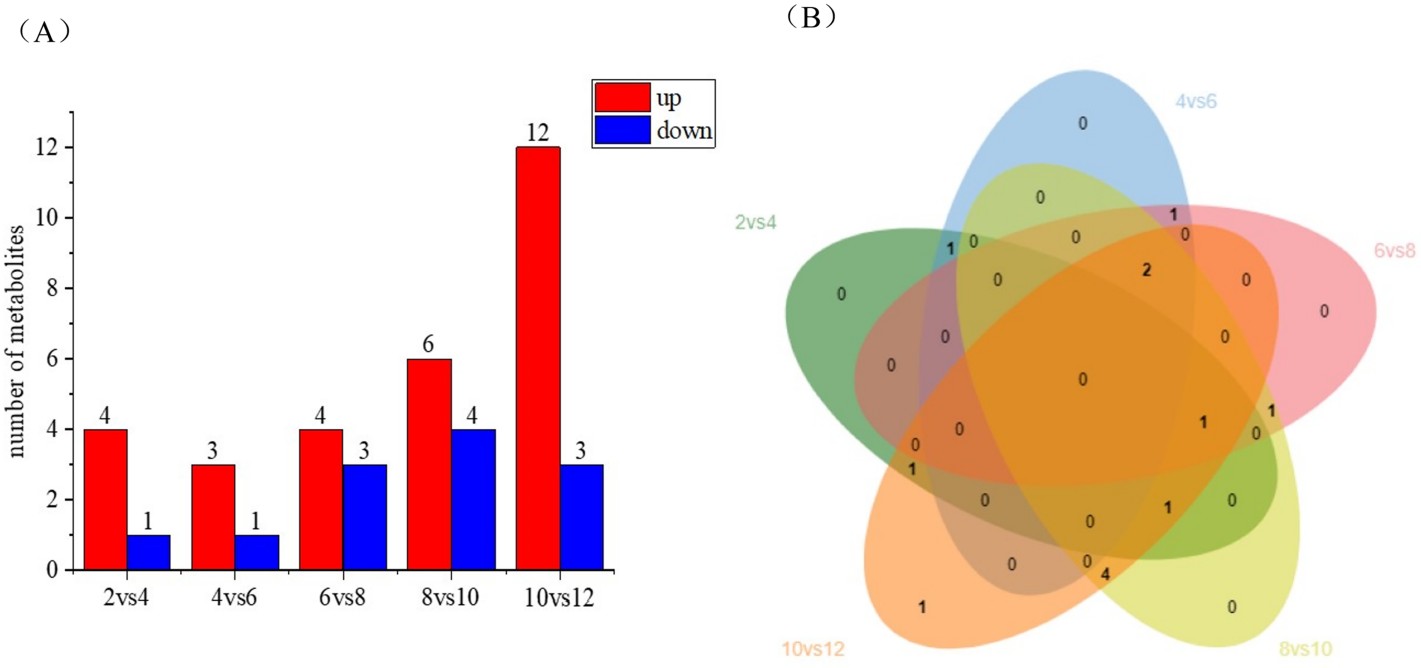

**Figure 5 Differential metabolite statistics of local coconut.** (A) Histogram of differential metabolites (B) Venn diagram of differential metabolites; By clustering the data, patterns and similarities in lignan metabolite profiles, helping to uncover potential relationships between lignan production and the fruit's developmental processes.

metabolites. As shown in Fig. 5B, the number of differential metabolites decreased during the first 2–6 months of development, with up-regulated metabolites being more prominent. Between 6–12 months of development, the number of differential metabolites increased, while up-regulated metabolites became more abundant. Notably, there are fewer metabolites early in coconut water development, and more metabolites later in development.

The study analyzed lignan metabolite in coconut water across six developmental stages of coconut (Fig. 6) and divided 13 species into two groups following Z-score normalization treatment. In the first group, there were 11 species of lignans, and the content of Rohan rosin-4,4′-di-o-glucoside, Cephaloethin, Rohan rosin phenol-4′-O-glucoside decreased from 2 to 4 months after coconut water development. The content of cefaloethin was highest at 2 months of age in local coconut water, while the content of Rohan Rosin Phenol-4′O-Glucoside was highest in the coconut water of Wenye No. 5.

At 4 months, Rohan rosin-4,4′-di-o-glucoside content was highest, while Wenye No. 5 coconut water had high levels of Epiturinol, Turbinol, turbinol-4-O-glucoside, and Isobarinin-9′-o-glucoside at 10 months. At 12 months, coconut water showed significant levels of 5′-methoxy-rohanoside, cycloisoperinolin-4-O-glucoside, and Larinin-4′-o-glucoside, with Wenye No. 5 being higher than local coconut water. In group 2, the highest

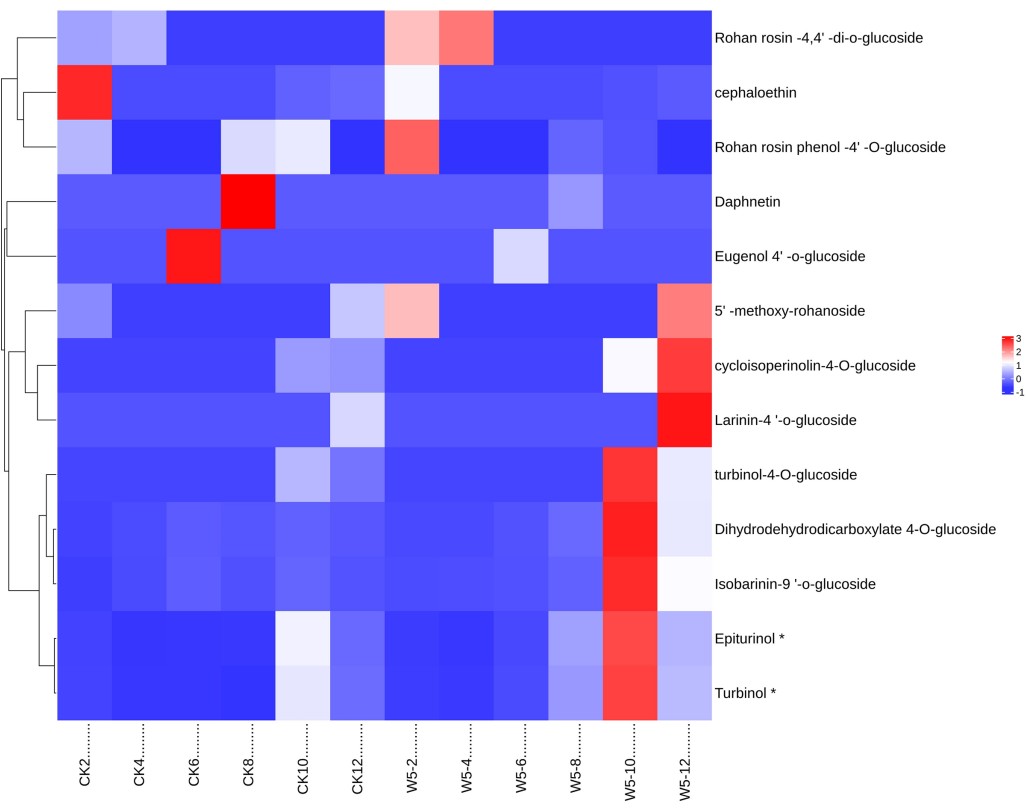

**Figure 6 Cluster heat map of different lignan metabolites in coconut water at different developmental stages.** (A) Histogram of differential metabolites; (B) Venn diagram of differential metabolites.

content of Eugenol 4′-o-glucoside in local coconut water was observed at 6 months of development, while Daphnetin content was highest at 8 months.

**Note:** CK-2: Local coconut develops for 2 months; CK-4: Local coconut develops for 4 months; CK-6: Local coconut develops for 6 months; CK-8: Local coconut develops at 8 months; CK-10: Local coconut develops for 10 months; CK-12: Local coconut develops for 12 months; W5-2: Wenye No. 5 develops for 2 months; W5-4: Wenye No. 5 develops for 4 months; W5-6: Wenye No. 5 develops for 6 months; W5-8: Wenye No. 5 develops for 8 months; W5-10: Wenye No. 5 develops for 10 months; W5-12: Wenye No. 5 develops for 12 months.

## Analysis of coconut CT image data
### CT numerical variance analysis of Wenye No. 5 and local coconut at different developmental stages

The CT values of different coconut components, such as coconut fiber, coconut shell, coconut flesh, and coconut water, showed significant variation across different coconut seasons (Table 1). The CT value of Wenye No. 5 coconut fiber significantly increased after 2 months of fruit age compared to 10 months of fruit age, indicating a 59.4% decrease in quality. This indicates that the fiber at 2 months of fruit age is of higher quality and more desirable compared to the fiber at 10 months of fruit age. The CT value of the coconut shell

**Table 1 Variances of growth indexes between Wenye No. 5 and local coconut at different developmental stages.**

| Developmental period | Variety | Coconut fibre (HU) | Coconut shell (HU) | Coconut meat (HU) | Coconut water (HU) |
|---|---|---|---|---|---|
| 2 | W5 | −13.17 ± 4.75a | 32.17 ± 1.67c | \ | 24.33 ± 1.17a |
| | CK | 0.80 ± 5.62a | 28.40 ± 2.33b | \ | 21.00 ± 0.96b |
| 4 | W5 | −21.22 ± 4.96a | 29.33 ± 3.21c | \ | 19.22 ± 1.82b |
| | CK | −15.30 ± 14.24a | 16.70 ± 1.57b | \ | 20.40 ± 0.94b |
| 6 | W5 | −70.67 ± 2.85b | 132.67 ± 5.04b | 47.83 ± 4.33a | 20.00 ± 0.50b |
| | CK | −86.80 ± 16.37b | 35.10 ± 19.27b | \ | 21.00 ± 2.74b |
| 8 | W5 | −67.83 ± 13.06b | 122.50 ± 6.51b | 45.83 ± 4.78a | 27.00 ± 1.76a |
| | CK | −760.40 ± 27.68c | 218.90 ± 2.78a | 10.30 ± 1.32b | 25.00 ± 0.52ab |
| 10 | W5 | −795.33 ± 23.32c | 201.83 ± 21.51a | 39.33 ± 0.60a | 25.50 ± 0.87a |
| | CK | −810.70 ± 20.63cd | 229.60 ± 4.47a | 12.20 ± 1.37b | 27.70 ± 1.97a |
| 12 | W5 | −766.34 ± 7.26c | 142.00 ± 8.92b | 35.77 ± 6.78a | 18.78 ± 1.25b |
| | CK | −864.50 ± 11.05d | 214.80 ± 13.79a | 16.70 ± 1.37a | 23.90 ± 1.91ab |

Note:
Values in the table are mean ± standard error, and different lowercase letters in the same column indicate significant differences ($P < 0.05$).

at 10 months of fruit age was significantly higher than that at 4 months. The value of coconut meat showed no significant difference between 2 to 4 months, with the CT value gradually decreasing after 4 months. Between 6 to 12 months, there was a slight decrease of 0.5% in the overall value. Interestingly, at 8-month, coconut water stood out with a higher CT value compared to other months. The study revealed an interesting finding regarding the local coconut fiber. The fiber content at 2 months of fruit age was significantly higher compared to that at 12 months, with a notable decrease in CT value of 1,081.6% The coconut shell at 10 months of fruit age exhibited a significant increase in CT value compared to the coconut shell at 4 months, with a 12.7% rise in CT value, indicating a notable difference in the maturity and development of the coconut shell over time. However, the CT value of coconut water showed a gradual increase after 8 months, with the CT value at 10 months significantly higher than that at 4 months.

In summary, this indicates that Wenye No. 5 coconut fiber and shell have superior properties compared to local coconut.

During the study, significant differences in CT values were observed between local coconut and Wenye No. 5 at different fruit ages. The most notable disparities occurred at 6, 8, 10 and 12 months (Fig. 7). These findings suggest that the fruit age plays a crucial role in determining the CT values of the coconuts, with these months showing the most distinct variations. The difference in CT values between the local coconut fruit age sample and Wenye No. 5 in 8 months was primarily reflected in pc2. In this analysis, the variable with the highest contribution to pc2 was fiber. The secondary factor contributing to the difference between the two varieties of coconut at 8 months was the coconut water. Similarly, the main factor contributing to the difference between the two coconut varieties at 12 months fruit age is the composition of the coconut meat and coconut shell. There is a strong positive correlation between coconut meat and coconut shell indicating that as the amount of coconut meat increases, the amount of coconut shell also tends to increase.

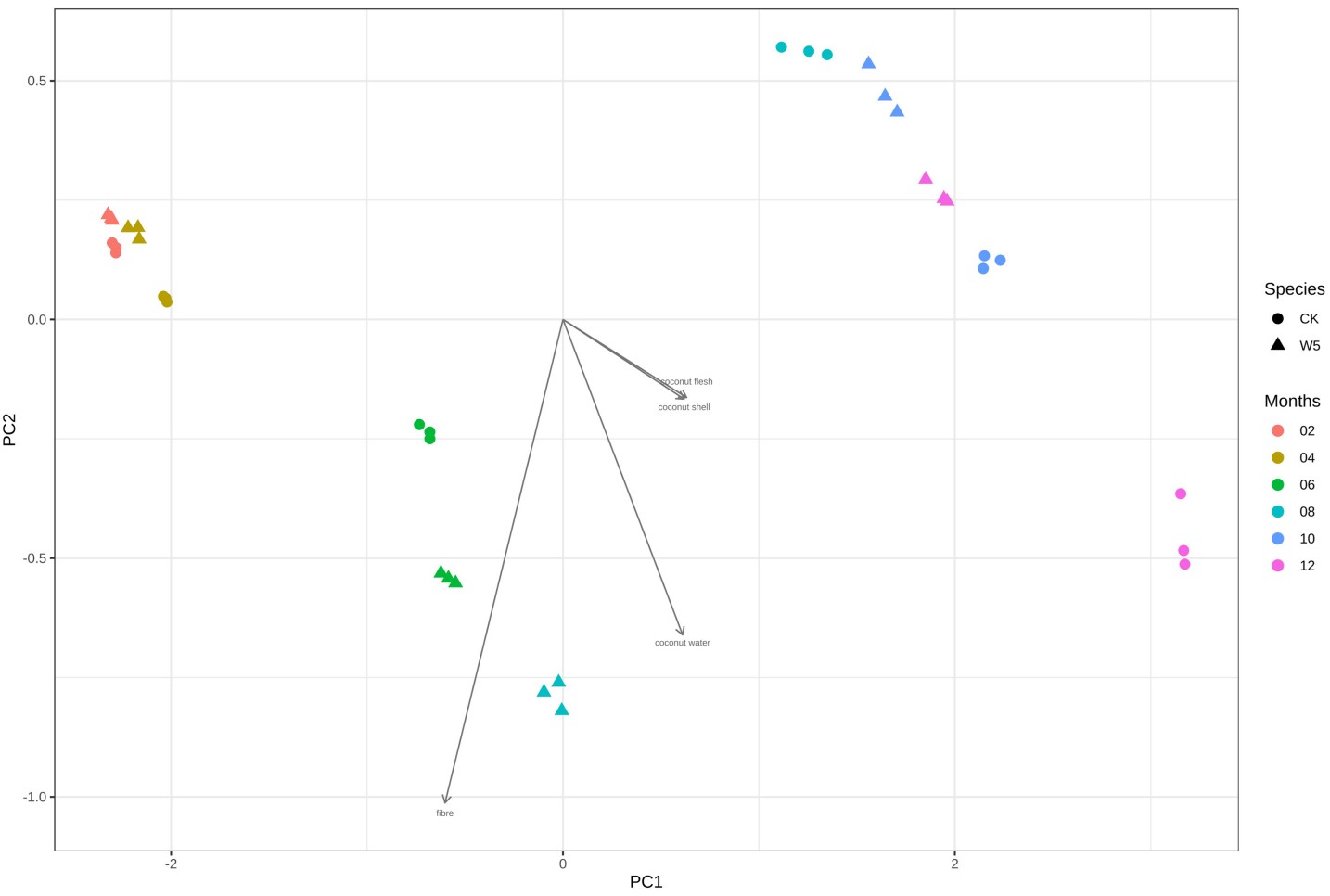

**Figure 7 Difference analysis of CT values of different coconut varieties in different periods.**

### Correlation analysis of CT values between Wenye No. 5 and local coconut at different developmental stages

According to Table 2, a significant positive correlation was observed between coconut meat and coconut shell of Wenye No. 5 ($P < 0.05$). However, no significant correlations were found between the other indicators. Additionally, local coconut water showed a positive correlation with coconut shell ($P < 0.01$), but no significant correlation was found between other indexes.

On the whole, as the fruit age increased, the coconut shell and coconut meat of Wenye No. 5 were significantly positively correlated, indicating that coconut meat increased with the growth of coconut shell.

### Principal component analysis of CT values of Wenye No. 5 and local coconut at different developmental stages

As shown in Table 3, the cumulative contribution rate of the first two principal components (eigenvalue > 1) in Wenye No. 5 reached 85.72%.This means that the first two principal components can represent 85.72% of the information in all resources. In the first

**Table 2 Correlation between growth indexes of Wenye No. 5 and local coconut at different developmental stages.**

| Variety | Growth index | Coconut fibre | Coconut shell | Coconut meat | Coconut water |
|---|---|---|---|---|---|
| W5 | Coconut fibre | 1 | −0.76* | −0.39 | 0.05 |
|  | Coconut shell | −0.76* | 1 | 0.86* | 0.23 |
|  | Coconut meat | −0.39 | 0.86* | 1 | 0.19 |
|  | Coconut water | 0.05 | 0.23 | 0.19 | 1 |
| CK | Coconut fibre | 1 | −0.99** | −0.55 | −0.89** |
|  | Coconut shell | −0.99** | 1 | 0.52 | 0.92** |
|  | Coconut meat | −0.55 | 0.52 | 1 | 0.17 |
|  | Coconut water | −0.89** | 0.92** | 0.17 | 1 |

Note:
* indicates significant difference ($P < 0.05$).
** indicates extremely significant difference ($P < 0.01$).

**Table 3 Principal components of different tissues in different developmental stages of coconut.**

| | Variety | Principal component 1 | Principal component 2 |
|---|---|---|---|
| Coconut fibre | W5 | −0.50 | 0.40 |
|  | CK | −0.56 | 0.02 |
| Coconut shell | W5 | 0.64 | −0.02 |
|  | CK | 0.56 | −0.07 |
| Coconut meat | W5 | 0.56 | 0.11 |
|  | CK | 0.33 | 0.88 |
| Coconut water | W5 | 0.17 | 0.91 |
|  | CK | 0.51 | −0.47 |
| Eigenvalue | W5 | 2.39 | 1.04 |
|  | CK | 3.12 | 0.85 |
| Percentage (%) | W5 | 59.72 | 26.00 |
|  | CK | 77.98 | 21.36 |
| Cumulative percentage (%) | W5 | 59.72 | 85.72 |
|  | CK | 77.98 | 99.35 |

principal component, the coconut shell demonstrates a higher load and a positive value, indicating a strong influence on the overall variability of the data in this component. Conversely, the load of coconut water on the second principal component is also higher and positive; suggesting that coconut water significantly contributes to explaining the variation in the data along the second component. The amount of information provided by the first and second principal components b was 59.72% and 26.00%, respectively. This cumulative contribution rate of 85.72% suggests that these two components can effectively represent a significant portion of the information about developmental traits, indicating that principal component analysis can effectively reduce the original information into the principal component.

In conclusion, as development days increase, there are clear disparities in the principal component analysis of fruit growth indices between Wenye No. 5 and local coconut varieties. In principal component 1, the Wenye No. 5 coconuts exhibits higher levels of coconut fiber, coconut shell, and coconut meat compared to the local coconut variety. Notably, the load value of coconut shell is the highest in principal component 1 for both varieties, suggesting that the coconut shell plays a significant role in differentiating between the Wenye No. 5 and local coconut varieties in terms of their composition. Principal component 2 plays a significant role in distinguishing between coconut fiber, coconut shell, coconut water, and the characteristic values of Wenye No. 5 compared to local coconut varieties. The values of these components were observed to be higher in Wenye No. 5, indicating unique qualities and attributes in terms of fiber, shell, and water content.

## Assessment of the association between metabolites and CT values in different parts of coconut

The order of the effects of coconut meat on all metabolites is as follows (Fig. 8): Rohan rosin-4,4′-di-o-glucoside > Cephaloethin > Rohan rosin phenol -4′-O-glucoside > Eugenol 4′-o-glucoside > 5′-methoxy-rohanoside > Dihydrodehydrodicarboxylate-4-O-glucoside ≈ Isobarinin-9′-o-glucoside > Daphnetin > turbinol-4-O-glucoside > Turbinol > Epiturinol > cycloisoperinolin-4-O-glucoside ≈ Larinin-4′-o-glucoside Similarly, coconut water has the greatest impact on Daphnetin; both coconut shell and coconut fiber had the greatest effect on Larinin-4′-o-glucoside.

**Note:** In Fig. 8, arrows indicate growth indicators; The direction pointed by the arrow indicates the change trend of the growth index; The vertical projection of metabolites on the arrow and its rays can determine the influence of a certain growth index on all metabolites.

## DISCUSSION

The study revealed significant differences in lignan metabolite content in coconut fruit at different developmental stages ($P < 0.05$), with a notable trend of change as the coconut developed. The local coconut variety showed higher levels of Eugenol 4′-o-glucoside at 6 months of fruit age. Additionally, at 8 months of fruit age, the local coconut variety exhibited higher levels of Daphnetin, compared to Wenye No. 5. The study found that the levels of Cephaloethin and Rohan rosin phenol-4′-O-glucoside in coconut water from 10 to 12 months were higher compared to Wenye No. 5. However, the concentrations of other lignans decreased during this period. The variations in lignan metabolites over different phases of development suggest a pattern of change. Initially, there is a decrease in these metabolites, followed by an increase, and ultimately a decline, with the highest levels observed around 12 months. This trend indicates that lignan metabolism undergoes dynamic changes throughout development. This is differ from previous results, which showed that the two lignans present in each part of the Taoerqi grown plant formed a W shape with seasonal variations, with maximum levels occurring 7 months after harvest (Li et al., 2015).

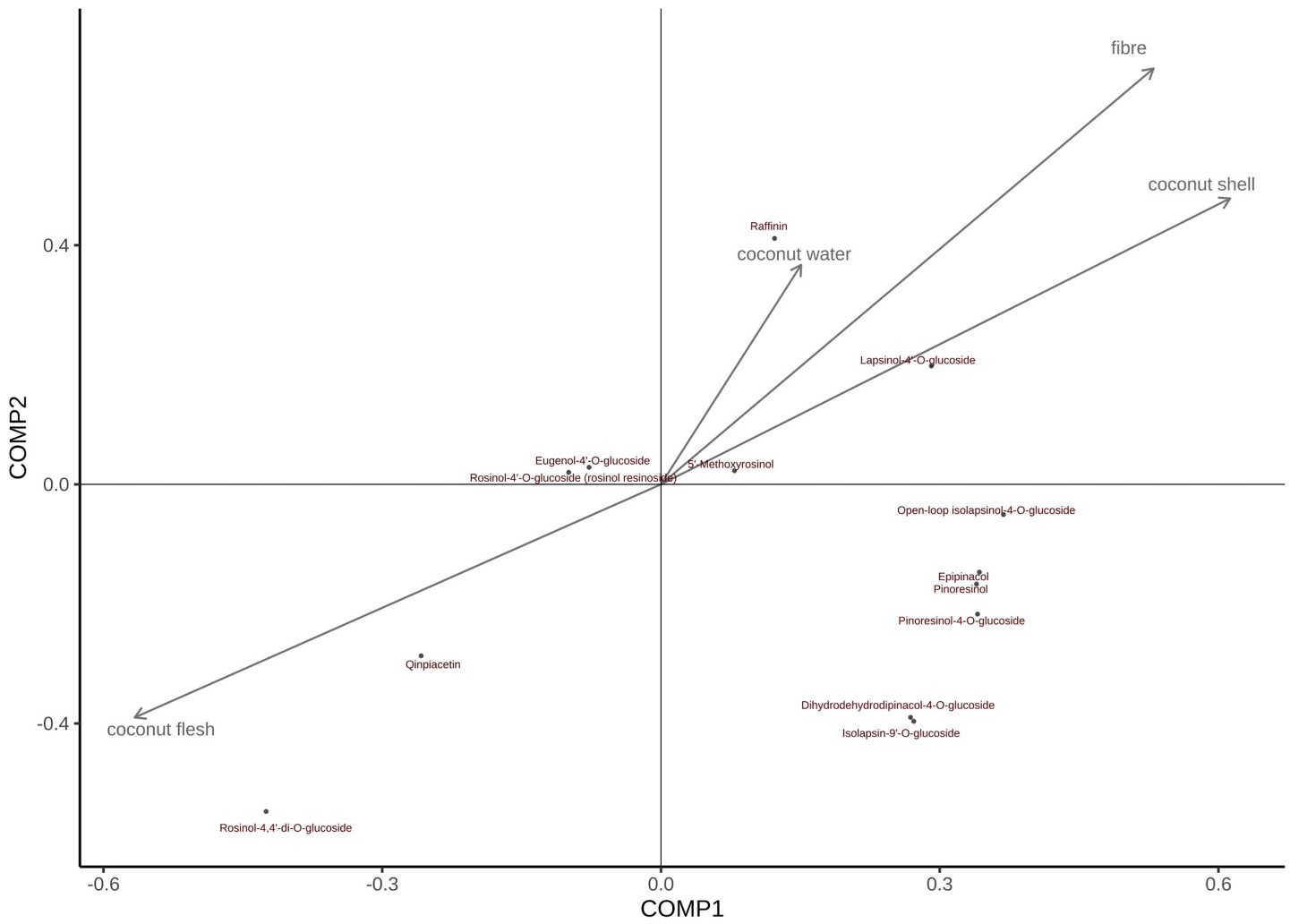

**Figure 8 PLS was used to assess the relative importance of metabolites and CT values in different tissues of coconut.**

In contrast, a study conducted by *Zhang et al. (2019)* found that the dried bark of the Euplia Ulmoides plant had a significant amount of lignan, and this content increased as the plant aged. The results suggest that the variance in findings could be due to the different varieties of the plant that were tested.

The levels of turbinol-4-O-glucoside in coconut water were found to be highest at the 10-month, while the highest levels of 5′-methoxy-rohanoside were observed at 12 months. This may be attributed to differences in secondary metabolite synthases and pathways, which can lead to variations in lignan synthesis and accumulation at different developmental stages. The inconsistent peak and trough periods of different lignans in coconut water indicate that various metabolites follow distinct synthesis and accumulation patterns within the same plant part (*Dong et al., 2005*). This suggests that the production and storage of lignans in coconut water are not uniform and may be influenced by different factors. Understanding these variations can provide valuable insights into the metabolic

processes occurring within the coconut plant and potentially lead to the identification of specific enzymes or regulatory mechanisms involved in lignan synthesis.

PLS analysis revealed significant difference in the CT values of local coconut and Wenye No. 5 at 6, 8, 10, and 12 months. The variable with the largest contribution to pc2 between local coconut and Wenye No. 5 at 8 months of fruit age was fiber. The growth difference between the two varieties at this age may be the main factor contributing to the observed variation. Additionally, coconut water could be a secondary factor influencing this difference. It is essential to consider both factors when analyzing the variations between the two varieties. At 12 months of fruit age, the main factors leading to the difference between the two varieties may be coconut meat and coconut shell. The study shows a strong positive correlation between coconut meat and coconut shell, suggesting that as the amount of coconut meat increases, the amount of coconut shell also increases. On the other hand, the correlation between fiber and coconut meat was found to be weak, indicating no clear relationship between the amount of fiber and the amount of coconut meat. However, the study found that the content of lignans in two kinds of rattan was closely related to habitat factors and growth traits and positively correlated with vine stem length, vine stem diameter and leaf length (*Zhang, Lv & Chen, 2021*), although, none of these correlations were found to be statistically significant. Coconut meat found to have the greatest effect on Rohan rosin-4,4′-di-o-glucoside. On the other hand, coconut water has a notable effect on Daphnetin. Additionally, coconut shell and coconut fiber have strong influence on Larinin-4′-o-glucoside. Currently, there is limited research on lignans in coconut water. However, it is worth noting that Tartary buckwheat root primarily contains 7-hydroxyresinol, while flaxseed is known to contain diisorietalinol (*Pexová Kalinová et al., 2022*; *Charlet et al., 2002*). These lignans have been studied for their potential health benefits and may play a role in various physiological processes. Further research is needed to fully understand the presence and effects of lignans in coconut water and their potential impact on human health.

## CONCLUSIONS

This preliminary discussion on the correlation between coconut fruit non-destructive testing and coconut fruit development process presents a fresh perspective and approach for future research in this field. By exploring the relationship between non-destructive testing methods and the development process of coconut fruits, new insights and understandings can be gained. CT-based non-destructive methods offer significant advantages over traditional manual and destructive methods for breeding crops with complex genetic makeup or identifying genetic mutations associated with specific traits. Studies on maize kernels (*Li et al., 2023*), barley spikes (*Ling et al., 2023*), coconut fruits (*Yu et al., 2022*), passion fruit (*Lu et al., 2023*) demonstrate that CT imaging combined with deep learning models enables high-throughput, accurate, and detailed phenotypic trait analysis.

The study analyzed lignan metabolites and CT image data of coconut fruits at different stages between Wenye No. 5 and local coconut. Results showed that the total lignan content in coconut water followed a specific pattern throughout the fruit's development.

Initially, the lignan content decreased, indicating a potential loss of these compounds. However, as the fruit continued to develop, the lignan content started to increase, suggesting that lignans were being synthesized or accumulated. After reaching maturity, the lignan content gradually decreased again, possibly due to degradation or other metabolic processes. The findings revealed a significant correlation between different parts of coconut fruit and lignan metabolites. This analysis can also help identify the optimal developmental stage for harvesting coconut water to maximize its lignan metabolite content. Such knowledge is crucial for industries involved in the production and commercialization of coconut water products.

Furthermore, the CT imaging method provided precise measurements of various traits such as volume, surface area, and internal structures, allowing for the identification of genetic markers associated with key traits. The non-destructive nature of CT imaging ensured that spatial information was retained, enhancing the understanding of genetic mechanisms underlying complex traits. This facilitates more efficient breeding strategies and genetic mutation identification.

Overall, investigating the changes in lignan metabolite content in coconut water at different developmental stages is a promising avenue for further research and development in the field of agricultural science.

### Funding

This work was supported by the Central Finance Forestry Science and Technology Promotion Demonstration Project (Qiong [2021]TG 05) and the International Exchange and Cooperation in Agriculture "Belt and Road" Tropical Agricultural Resources and Technology Cooperation (2001021). The APC was supported by the Forestry Science and Technology Promotion and Demonstration Fund (other promotion and demonstration) project of the central government, grant number Qiong [2024] No. TG07. The funders had no role in study design, data collection and analysis, decision to publish, or preparation of the manuscript.

### Grant Disclosures

The following grant information was disclosed by the authors:
Central Finance Forestry Science and Technology Promotion Demonstration Project: Qiong [2021]TG 05.
International Exchange and Cooperation in Agriculture "Belt and Road" Tropical Agricultural Resources and Technology Cooperation: 2001021.
Forestry Science and Technology Promotion and Demonstration Fund (other promotion and demonstration) project of the central government:  Qiong [2024] No. TG07.

### Competing Interests

The authors declare that they have no competing interests.

## Author Contributions

- Chengxu Sun performed the experiments, prepared figures and/or tables, and approved the final draft.
- Xuejing Ma performed the experiments, prepared figures and/or tables, and approved the final draft.
- JeromeJeyakumar John Martin performed the experiments, authored or reviewed drafts of the article, and approved the final draft.
- Hongxing Cao conceived and designed the experiments, authored or reviewed drafts of the article, and approved the final draft.
- Yu Zhang analyzed the data, prepared figures and/or tables, authored or reviewed drafts of the article, and approved the final draft.
- Yanming Gao analyzed the data, prepared figures and/or tables, and approved the final draft.
- Chunyu Xing analyzed the data, prepared figures and/or tables, authored or reviewed drafts of the article, and approved the final draft.
- Mingming Hou analyzed the data, authored or reviewed drafts of the article, and approved the final draft.

## Data Availability

The raw measurements are available in the Supplemental File.

## Supplemental Information

Supplemental information for this article can be found online at http://dx.doi.org/10.7717/peerj.18049#supplemental-information.

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
