# Peer review of "Preliminary study on the association between lignan metabolites and CT non-destructive testing of coconut fruit at different developmental stages"

_PeerJ, doi:10.7717/peerj.18049_

## Round 0.1 · original submission · Major Revisions

Please address concerns of the reviewers and amend manuscript accordingly.

·

Basic reporting

The current study aimed at finding out association between lignan metabolites and CT non-destructive testing of coconut fruit at different developmental stages. They have chosen two different varieties of coconut to find differences in metabolites. Figures are of importance but need to improve its pixels for the publication purpose. Authors should supply high quality of pics for publications. A through English proof reading is required.

Experimental design

The experiment is well planned and executed. Recording of data and analysis are proper. They have standard method of liquid chromatography to measure the levels of lignan metabolites in coconut water.

Validity of the findings

The experiment opens up new insight on the use of non-destructive method (CT-based). However it should correlated with its practical utility for example selection of a particular variety of plant after some breeding experiments.

Additional comments

Authors should should put forth some suggestions in what are the utility of this work in agricultural sciences for exam to select and reject some plant varieties.

Reviewer 2 ·

Basic reporting

1. The text is too small in FIgure 4-8.
2. What are “up” and “down” labels in Figure 5A?
3. Figure 5B is completely unreadable.
4. The color bar in Figure 6 is too small to read.
5. Significant improvements in English writing are required.
6. What are a,b,c,d in asscoaited with data shown in Table 1?
7. What is x-axis in Figure 1-3?

Experimental design

1. What is the purpose of focusing on 2, 4, 6, 8, 10, 12 months data? Why not having a more fine grained study (e.g., 1,2,3,4,5,6,7,8,9,10,11,12 months)?
2. The authors failed to give enough introduction of Cephaloethin, Epiturinol, Turbinol, turbinol-4-O-glucoside, Rohan rosin phenol -4’-O-glucoside, 5’-methoxy-rohanoside, Isobarinin-9’-o-glucoside and Larinin-4’-o-glucoside. These have been frequently discussed in the manuscript without enough introduction to help readers understand the point of these measurements.
3. It is also unclear why the authors did the comparison between Wenye No.5 coconut and local coconut. Without clarifying enough rationale, almost 50% of the results are meaningless.

Validity of the findings

1. The results discussed between line 248 and 269 do not support the conclusion that Wenye No.5 coconut fiber and shell have superior properties compared to local coconut.

---

## Round 0.2 · accepted · Accept

All the issues pointed by the reviewers were addressed and the revised manuscript is acceptable now.

Reviewer 2 ·

Basic reporting

The authors have addressed all of my concerns.

Experimental design

The authors have addressed all of my concerns.

Validity of the findings

The authors have addressed all of my concerns.